# MyoChallenge 2024: A New Benchmark for Physiological Dexterity and Agility in Bionic Humans

**Cheryl Wang**[1*]   **Chun Kwang Tan**[2*]   **Balint Hodossy** [3*]   **Eric Lyu**[4*]
**Pierre Schumacher**[5*]   **James Heald**[6†]   **Kai Biegun**[6†]   **Samo Hromadka**[6†]   **Maneesh Sahani**[6†]
**Gunwoo Park**[7‡]   **Beomsoo Shin**[7‡]   **Jonghyun Park**[7‡]   **Seungbum Koo**[7‡]
**Chenhui Zuo**[8§]   **Chengtian Ma**[8§]   **Yanan Sui**[8§]   **Nicklas Hansen**[9¶]
**Stone Tao**[9¶]   **Yuan Gao**[9¶]   **Hao Su**[9¶]   **Seungmoon Song**[2]   **Letizia Gionfrida** [4]
**Massimo Sartori**[10]   **Guillaume Durandau**[1]   **Vikash Kumar**[11]   **Vittorio Caggiano**[11]

[1]McGill University, Canada    [2]Northeastern University, USA
[3] Imperial College London, UK [4] King's College London, UK
[5]Max Planck Institute for Intelligent Systems, Germany [6] University College London, UK
[7]Korea Advanced Institute of Science and Technology, South Korea
[8] Tsinghua University, China    [9] University of California, San Diego, USA
[10]University of Twente, Netherlands [11]MyoLab, USA

[†]Team Muscle Heads    [‡]Team MSKBioDyn    [§]Team LNS Group    [¶]Team Loco UCSD

## Abstract

Recent advancements in bionic prosthetic technology offer transformative opportunities to restore mobility and functionality for individuals with missing limbs. Users of bionic limbs, or bionic humans, learn to seamlessly integrate prosthetic extensions into their motor repertoire, regaining critical motor abilities. The remarkable movement generalization and environmental adaptability demonstrated by these individuals highlight motor intelligence capabilities unmatched by current artificial intelligence systems. Addressing these limitations, *MyoChallenge*'24 at NeurIPS 2024 established a benchmark for human-robot coordination with an emphasis on joint control of both biological and mechanical limbs. The competition featured two distinct tracks: a manipulation task utilizing the myoMPL model, integrating a virtual biological arm and the Modular Prosthetic Limb (MPL) for a passover task; and a locomotion task using the novel myoOSL model, combining a bilateral virtual biological leg with a trans-femoral amputation and the Open Source Leg (OSL) to navigate varied terrains. Marking the third iteration of the MyoChallenge, the event attracted over 50 teams with more than 290 submissions all around the globe, with diverse participants ranging from independent researchers to high school students. The competition facilitated the development of several state-of-the-art control algorithms for bionic musculoskeletal systems, leveraging techniques such as imitation learning, muscle synergy, and model-based reinforcement learning that significantly surpassed our proposed baseline performance by a factor of 10. By providing the open-source simulation framework of MyoSuite, standardized tasks, and physiologically realistic models, MyoChallenge serves as a reproducible testbed and benchmark for bridging ML and biomechanics. The competition website is featured here: https://sites.google.com/view/myosuite/myochallenge/myochallenge-2024.

---

[*]co-first

# 1 Introduction

One hallmark of human motor intelligence is the remarkable capacity to interact dynamically and adaptively with the environment. This capability becomes even more extraordinary when trauma or disease compromises our motor system, and we successfully recover or augment motor functions through artificial bionic systems. The resilience, dynamics and adaptability demonstrated by the human motor system under these conditions highlight the critical need for accurate modeling approaches that include not only typical motor functions but also the complexities of adaptation and recovery [1]. Incorporating these adaptive aspects into biomechanical modeling frameworks promises significant advancements in rehabilitation strategies, prosthetics, and assistive technologies, ultimately improving the quality of life for individuals facing motor impairments [2, 3].

While numerous simulation platforms and models have been proposed, existing systems often lack the scalability and realism required, leaving the control of integrated musculoskeletal and bionic systems an open challenge [4, 5, 6]. A key limitation is the inability of current neuromechanical models, despite their utility in testing control theories and simulating biologically realistic motion, to adapt coherently across tasks such as manipulation and locomotion. Hence, a comprehensive musculoskeletal bionic simulation environment would provide a robust testbed, enabling cross-disciplinary collaboration in robotics, motor control, physiology, and machine learning to develop and validate control strategies with substantial real-world implications [7, 8, 9].

Recent years have witnessed remarkable progress in biomechanics, machine learning [10, 11, 12], neuroscience, assistive devices [13, 14, 15], and physics-based simulation platforms [16, 17, 18, 19]. In computational biomechanics, several benchmarks have emerged over the past decade. The osim-rl NIPS 2018 Challenge: AI for Prosthetics [20] focused exclusively on lower-limb prostheses for walking, while other competitions [21, 22, 23, 24] benchmarked human motor control but neglected the integration of prosthetics with human biomechanics. Currently, no public benchmark exists that combines high-fidelity digital assistive models, advanced control algorithms, and modern learning architectures. Developing such a benchmark would provide critical insights into human-prosthesis coordination.

To address this critical gap, we introduce MyoChallenge'24, a competition designed to establish novel benchmarks for dexterous manipulation using upper-limb prosthetics and agile locomotion with simulated lower-limb amputations. The competition platform advances the development of realistic biomechanical digital twins of human amputees and seeks to answer: *Can we achieve human-level coordination between physiological digital twins and bionic prosthetic limbs?* The competition features two independent tracks focused explicitly on musculoskeletal simulations integrated with prosthetic limb co-control. This initiative aims to establish a realistic and transferable benchmark for bionic limb control strategies, advancing the state of rehabilitation and assistive technologies.

# 2 The MyoChallenge'24 Competition

In *MyoChallenge'24*, we present two tracks. The first track requires bi-manual coordination of the myoMPL model – a combination of a virtual biological arm and a Modular Prosthetic Limb (MPL) [25]. A second track features a new myoOSL model made from the combination of a virtual bilateral biological leg with a trans-femoral amputation together with an Open Source Leg (OSL) [13]. The competition was divided into two phases, a preparation phase and a submission phase. During the first stage, two main environments were open-sourced for participants to develop and test out early solutions. The second phase introduces variation into the environment parameters and opens up submission to the EvalAI platform. To promote diversity in science, a special DEI award for participants from an underrepresented population and a Student Award to promote participation among undergraduate students/High School.

The tasks and environment are available by cloning the MyoChallenge24 GitHub template (`https://github.com/MyoHub/myochallenge_2024eval`). The EvalAI platform (`https://eval.ai`) was used for hosting the challenge and to run the evaluation. Participants were asked to upload their behavior policies to Eval AI, which automatically evaluated them and updated results on a scoreboard. Final scores were averaged over 100 trials of multiple seeds and unseen task variations. The competition environment would be continuously available within the myosuite repository to encourage further usage and development in bionic limb research. In the following sections, we

present the simulator and musculoskeletal models (Sec. 2.1), and the proposed tasks and evaluation metrics (Sec. 2.2).

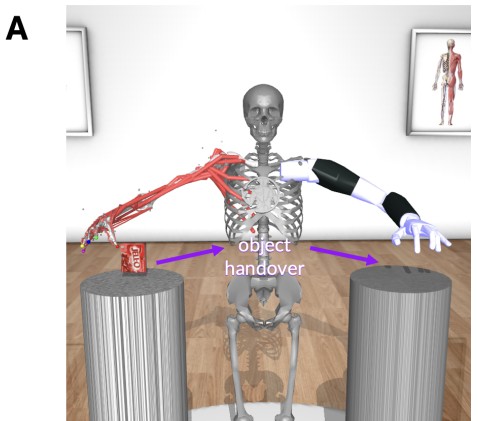
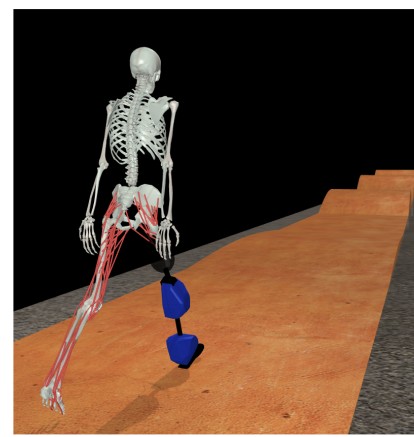

Figure 1: Two tracks of MyoChallenge 2024. **A.** the manipulation track where an object needs to be relocated via a handover, necessitating the coordination of a musculoskeletal and a robotic prosthetic arm. **B.** the locomotion track, where a locomotion agent must traverse diverse terrain while walking with one musculoskeletal and one prosthetic leg.

## 2.1 Simulation Framework and Musculoskeletal Models

**Simulation Framework** The competition and the full set of available musculoskeletal models are embedded within MyoSuite, an open-source collection of environments and tasks that runs in the MuJoCo physics engine [26]. MyoSuite provides physiologically realistic, musculoskeletal full-hand models [27] within a simulation framework that significantly surpasses the speed of state-of-the-art musculoskeletal simulators [28, 29] used in earlier competitions—achieving speed-ups of up to 4000 times (see Figure 7 in [19]). Additionally, MyoSuite supports comprehensive contact dynamics, a crucial feature absent in most competing platforms, enabling the simulation of complex, contact-rich manipulation behaviors. Specifically in MuJoCo, the muscles are modelled as actuators attached to tendons that are assumed to be infinitely stiff.

**MyoArm and MPL Musculoskeletal Model** The right arm of the myoArm model, comprising 27 degrees of freedom (DOFs) and 63 muscle-tendon units, is adopted from the OpenSim MoBL arm model [30, 31], converted via MyoConverter [28, 27]. This model has been utilized in prior MyoChallenges for manipulation tasks [23, 32]. The entire left arm is replaced by the MPL. Detailed information about the myoArm model is provided in Appendix A.2 and [33].

**MyoLeg and OSL Musculoskeletal Model** The myoLeg model is adapted to represent an individual with a right-leg transfemoral amputation at approximately 50% of femoral length. Initially comprising 28 DOFs and 80 muscle-tendon units, the original myoLeg model was derived from the OpenSim full-body model [34] using MyoConverter [27], and was previously featured in the MyoChallenge [32]. Adjustments to reflect the transfemoral amputation include removal of muscles controlling the knee and ankle joints on the amputated side [35], and recalibration of femoral inertial properties to account for altered geometry and mass distribution of the residual limb. Detailed information about the MyoLeg model can be found in Appendix A.2 and [33].

## 2.2 Tasks and Evaluations

### 2.2.1 Manipulation Track

**Task** The manipulation track (Fig.1-A) features a bimanual object relocation task, where the control of a musculoskeletal arm must be coordinated with that of a robotic prosthesis. A key source of complexity in this task is the handover that needs to take place between the two arms. The object is always cuboid in shape, but its exact dimensions are sampled from a distribution shown in Appendix A.3. The torso and lower limbs of the agent are fixed at the center of a circular stage, with two pillars positioned such that each arm can only access the pillar closest to it. The start and goal

positions are specified at the centers of the top surfaces of the left and right pillars. At the beginning of each simulation trial, both the object's initial and target positions are randomly initialized. The environmental variations are detailed in Appendix A.3 - Table 2.

**Observation** The controller receives a detailed, 222-dimensional observation vector describing the states of the body, object, and environment at every simulation timestep (Appendix A.3 - Table 3). This observation includes joint positions and velocities of the myoArm and the MPL, the 6-DOF position and velocity of the manipulated object, and muscle stimulation levels of the arm. Additionally, contact status is indicated by five binary labels specifying whether the object is in contact with the biological hand, MPL, start pillar, goal pillar, or any other environmental component. A comprehensive description of the observation space can be found in the MyoChallenge'24 documentation [36].

**Action** The action space is an 80-dimensional continuous vector, ranging from [-1, 1], which includes 63 muscle actions and 17 MPL position actions. These position actions correspond to the joint range of motion and are used for target angle-based control.

**Termination** A simulation trial is completed when the object is placed on the center of the goal pillar's face within 0.1 m along each axis. Each trial lasts up to 10 seconds and terminates early if the object falls below 0.3m. To ensure bimanual manipulation, each limb must maintain contact with the object for at least 1s.

**Evaluation Metric** The participants were first ranked based on the number of successful passes over to the goal pillar based on the previous termination condition. Teams that achieved at least a 90% success rate in a manipulation task qualified for further ranking in the time of completion. For teams whose scores are within 10% of each other, the third criterion of muscle activation effort determines their ranking, with less effort leading to a better ranking. Additionally, any solutions involving a peak contact force exceeding 1500N, indicative of 'throwing' rather than handover, are automatically labelled unsuccessful.

**Baseline Controller** We provided a benchmarking baseline policy that combined a simple position control for the MPL with a deep neural network, trained through reinforcement learning, to manage the grasping behavior of the myoArm. The baseline demonstrated a success rate of approximately 3% when the full object and environment variations were introduced.

### 2.2.2 Locomotion Track

**Task** The objective is to develop a controller for the amputee musculoskeletal model and to optimize OSL control selections to maximize walking distance across varied terrains, as illustrated in Fig.1-B. The environment consists of five 3 m wide and 100 m long terrains: 1) flat ground, 2) rough terrain, 3) hills, 4) stairs featuring three ascending and descending steps each, and 5) a composite of all terrain types (Fig. 2). The complexity of the terrain escalates with distance; at 100 m, roughness peaks at 60 cm, hills steepen to about 50 degrees with a height of 1.5 m, and each stair step measures 50 cm in height.

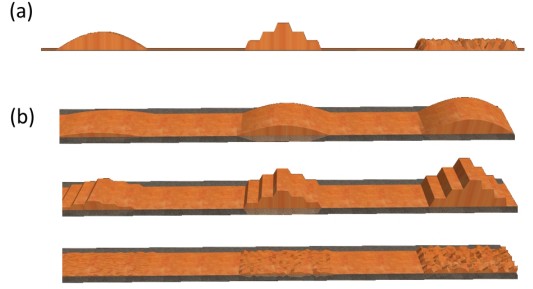

Figure 2: Terrains in myoOSL. (a) A side view illustrating an example of mixed terrain. (b) Various terrain types arranged with gradually increasing difficulty along the travel path.

**Observation** The controller/policy has access to proprioceptive data such as joint angles, velocities, muscle states, and a 10 x 10 height map of the 2 x 2 m area surrounding the amputee, updated at each time step (Appendix A.3 - Table 4).

**Action** The action space comprises 55 action inputs, where 54 are muscle actions, transformed into muscle stimulations, along with an additional output $[0, 1, 2, 3, 4]$, which allows for the selection among five predefined control parameter sets.

**Termination** A simulation trial is deemed complete when the amputee model successfully navigates the entire 100 m path without falling. Trials are terminated prematurely if the model falls (head height falls below 1.5 m) or strays outside the 3 m wide designated path.

**Evaluation Metrics**. Performance is first measured by the distance $D$ traveled in each episode, with higher distances receiving higher rankings. Teams must walk at least 20 meters to qualify for ranking with the time taken to complete the episode or until a fall occurs.

**Physiological Metrics** An additional physiological metric is used to encourage more naturalistic and realistic solutions. The participating teams must walk at least 20 meters to qualify for physiological ranking. The first criterion, pain, is assessed based on the average overextension torque at each joint in the MyoLeg. The second criterion, total muscle activation, is measured to estimate metabolic power with less effort leading to higher ranking.

**Baseline Controller** We do not currently provide a baseline controller for the myoOSL environment. However, baseline controllers for the intact myoLeg model that produce walking are available in MyoSuite [19]. These include the DEP-RL controller [12] and a reflex-based controller [37].

## 3   Results and Participation

This year's *MyoChallenge* had a total participation of 53 teams from over 15 countries. Across the three-month submission period, we had a total of 292 submissions and over 14,000 total downloads of MyoSuite. Among the teams that filled out the post-competition survey, 38% of the participating teams are composed of

Table 1: Submission results of MyoChallenge 2024

|  | Manipulation Track | Locomotion Track |
|---|---|---|
| 1st Place | 0.26 | 0.72 |
| 2nd Place | 0.18 | 0.19 |
| 3rd Place | 0.10 | N/A |

students only, with one team of high school students. Two of the top three winning teams for the manipulation track feature women researchers, a significant leap from the previous years' participation. However, no participants came from South America or Africa, highlighting a need to promote within these underrepresented communities.

Due to the complexity of each track's task and the need for familiarity with both biomechanics and robotics, only three teams in the manipulation track and two teams in the locomotion track were able to surpass the provided baselines (Table 1). The winning team in the manipulation track achieved the highest score of 0.26—a 44% increase on the next best score. In addition, each episode was completed in the least amount of time (5.9 seconds), using the least amount of muscular effort (0.034, muscle activation units), and applying the least peak contact force on the object (481 N). The winning team's model in the locomotion track was capable of walking on rough terrain up to 9 cm high, stairs with step heights up to 7 cm, and all hill terrains, including the highest hill at 24 cm with a maximum distance of 71.6 meters.

### 3.1   Manipulation Track

#### 3.1.1   First Place - Muscle Heads

**Architecture Overview** The Muscle Heads team used a combination of deep reinforcement learning and inverse kinematics in their winning solution to the manipulation track. To simplify learning, the task was broken down into a curriculum of subtasks. In stage 1 of the curriculum, the myoArm was trained to reach the object and grasp it. The grasped object was then moved to the robot hand, which was positioned in front of the agent using simple position control. In stage 2, the robot hand was trained to grasp the object as the myoHand released it. In stage 3, the robot hand was moved to the goal location (along with the grasped object) using inverse kinematics. The object was then released on the goal pillar by opening the fingers.

**Key Solution Insight - Muscle Synergies** To deal with the large number of muscles in the myoArm, rather than control each of the myoArm muscles individually, muscles were recruited in a coordinated manner using muscle synergies. The muscle synergies were learned by optimizing an objective that measures how much influence the agent can have over a task-relevant feature of the state known as a

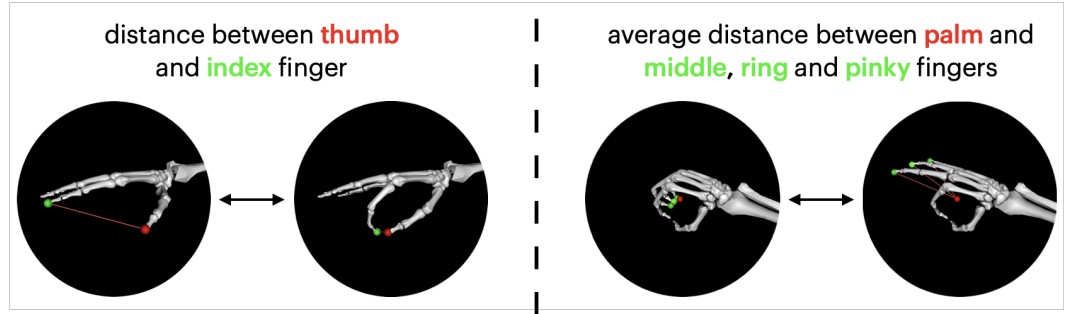

distance between **thumb**
and **index** finger

average distance between **palm** and
**middle**, **ring** and **pinky** fingers

Figure 3: Grasping variables controlled using muscle synergies. By controlling the distance between the tips of the thumb and index finger (left), a precision grip can be achieved and used to grasp the object. By controlling the average distance of the middle, ring, and pinky fingers from the palm (right), these fingers can be moved out of the way of the object, preventing them from interfering with the ability to grasp the object.

*controlled variable*. The controlled variable was defined as the concatenation of multiple variables that are key to reaching and grasping: the seven degrees of freedom of the shoulder, elbow and wrist joints (to control hand position and orientation); the distance between the tips of the thumb and index digits (to perform a precision grip, Figure - 3 left); and the average distance between the palm and the tips of the middle, ring and pinky fingers (to move these non-grasping fingers out of the way of the object, Figure - 3 right). In total, 9 synergies were used to coordinate the 63 muscles of the myoArm, as the controlled variable is 9-dimensional.

**Reward Design** In stage 1 of the training curriculum, the agent was rewarded for (i) bringing the myoHand thumb and index finger close to the object (with the condition that the thumb is behind the object and the index finger is in front of the object), (ii) bringing the object close to the midpoint of the robot thumb and index finger positions, and (iii) keeping the orientation of the object close to its initial orientation on the start pillar. In stage 2, the agent was rewarded for (i) bringing the robot thumb and index finger close to the object, and (ii) touching the object with the robot (with the condition that the robot thumb and index finger are on opposite sides of the object).

### 3.1.2 Second Place - LNS Group

**Architecture Overview** The LNS Group uses a combination of dynamical synergistic representation for myoArm and a trajectory interpolation for the MPL. The core idea is to reduce the control dimensionality by grouping functionally similar actuators into synergistic representations, while allowing state-dependent fine-tuning for individual actuators [38]. Their DynSyn method is designed to address the challenges of controlling high-dimensional and overactuated systems, such as the myoMPL.

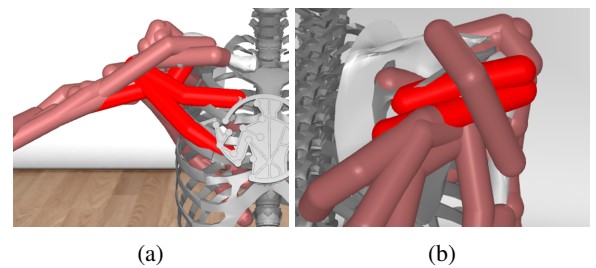

(a)  (b)

Figure 4: Muscle groups. (a) The middle and inferior fascicles of the pectoralis major and the coracobrachialis muscle are grouped together. (b) Infraspinatus and teres minor muscles are grouped together.

**Key Solution Insight - Dynamical Synergistic Representation** The DynSyn method begins by generating muscle length data through random joint velocity controls applied to the musculoskeletal model. These trajectories are segmented into shorter time intervals, and the correlation between length changes of each pair of muscles is calculated using cosine similarity. Based on the correlation matrix, the K-Medoids clustering algorithm is employed to group functionally similar actuators into synergistic bins. This process results in a reduced action space where each group of actuators shares a unified action. As shown in Fig. 4, the DynSyn algorithm successfully identified representative muscle-tendon groups in the myoArm model, reducing the control dimensionality while preserving functional synergy. The

middle/inferior pectoralis major and coracobrachialis were clustered. These muscles synergistically control shoulder adduction and rotation. Infraspinatus and teres minor form a group responsible for humeral external rotation. To simplify the control of the MPL and reduce the dimensionality of the action space, the LNS Group team designed four key postures that guide the prosthetic hand through the object transfer process. To ensure a natural and continuous movement, they interpolate between these postures over time, generating a smooth trajectory that facilitates stable object handling.

**Reward Design** The reward function is designed to guide the agent toward achieving the manipulation task while adhering to environmental constraints. As demonstrated in Appendix A.6 - Fig. 7, the reward function incorporates multiple components, each weighted to balance task objectives and stability. Each reward component serves a specific purpose: pre-grasp posture and lift bonus encourage the agent to achieve a stable initial grasp, while position distance and solved bonus drive goal-directed movement. Meanwhile, palm distance and shoulder elevation ensure smooth and stable interactions with the environment.

## 3.2 Locomotion Track

### 3.2.1 First Place - MSKBioDyn

**Architecture Overview** Team MSKBioDyn developed a multilayer perceptron controller trained using proximal policy optimization [39]. Curriculum learning was applied to facilitate task mastery by incrementally increasing difficulty. Initially, the policy learned to walk under low gravity, fixed initial conditions, flat terrain, and a high weight on style rewards. As proficiency improved, stronger gravity, variable initial conditions, steeper terrains, and increased weight on task rewards were progressively applied.

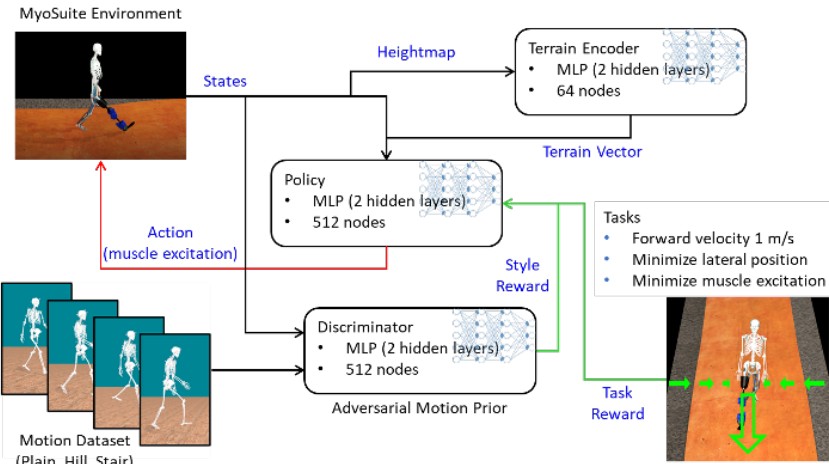

Figure 5: Training method of team MSKBioDyn. The policy network predicts optimal actions based on environment states, where the 10-by-10 heightmap is compressed into a 3-dimensional vector using the terrain encoder. The policy is trained using a style reward from AMP and a task reward from the environment.

**Key Solution Insight - Imitation Learning** The winning key of MSKBioDyn was to implement the adversarial motion prior (AMP) for imitation learning, which trains the policy with a discriminator [40]. As reference motions for AMP, the team used a comprehensive motion capture dataset previously collected by their group. This dataset contains full-body kinematics from 120 individuals walking on flat ground, stairs, and slopes, representing various walking styles and terrain conditions. Additionally, the observation dimensionality was reduced by excluding muscle lengths, velocities, and forces, and by compressing the heightmap into a three-dimensional terrain vector using a variational autoencoder. This autoencoder was pretrained on approximately 60,000 randomly sampled terrain observations from the track environment. The resulting terrain vector encoded orientation, scale, and positions of stairs and hills relative to the body.

**Reward Design** The policy was trained using a style reward from AMP (similarity to motions in the dataset) and a task reward Fig. 5. The task reward was maximized when the human model walked forward (y-axis) at 1 m/s without lateral (x-axis) displacement. Additionally, a squared excitation minimization term was included in the task reward to satisfy the physiological effort criteria.

### 3.2.2 Second Place - Loco UCSD

**Architecture Overview** The Loco UCSD team model the MyoChallenge locomotion problem as a reinforcement learning (RL) problem governed by a (non-terminal) Markov Decision Process. The aim is to learn a policy $\pi \colon \mathcal{S} \mapsto \mathcal{A}$ parameterized by a neural network that maximizes the discounted sum of rewards $\mathbb{E}_\pi \left[ \sum_{t=0}^T \gamma^t r_t \right]$, $r_t = \mathcal{R}(\mathbf{s}_t, \pi(\mathbf{s}_t))$ for an episode of length $T$, in expectation. While the policy $\pi$ can in principle be learned via any reinforcement learning algorithm, they base their approach upon the model-based reinforcement learning (MBRL) algorithm TD-MPC2 [41, 42] due to its strong empirical performance on high-dimensional control problems, including humanoid locomotion, as demonstrated in prior work [42, 43, 44].

**Key Solution Insight - MBRL** TD-MPC2 learns a (latent) world model from environment interactions and selects actions by planning with the learned model. All components of the world model are learned end-to-end using a combination of latent state prediction, reward prediction, and temporal difference losses, *without* ever decoding raw observations. TD-MPC2 is thus a *decoder-free* world model. During inference, TD-MPC2 plans actions via local trajectory optimization using (sampling-based) Model Predictive Path Integral (MPPI) [45]. To accelerate planning, TD-MPC2 additionally learns a policy prior using a maximum entropy RL objective in the latent space of the model; this policy prior is used to warm-start the sampling procedure. The TD-MPC2 world model architecture consists of the following five components, as detailed in Appendix A.8. They use TD-MPC2 as their choice of RL algorithm *without modification nor hyperparameter-tuning*, and instead focus on designing a reward function that is effective for RL.



Figure 6: **The TD-MPC2 architecture.** Observations $\mathbf{s}$ are encoded into their latent representation $\mathbf{z}$. The model then recurrently predicts actions $\hat{\mathbf{a}}$, rewards $\hat{r}$, and terminal values $\hat{q}$, *without* decoding future observations.

**Reward Design** Team Loco UCSD designs a reward function that has a total of 5 components:

$$\mathcal{R}(\mathbf{s}, \mathbf{a}) = \mathcal{R}_{\text{forward\_vel}}(\mathbf{s}) \cdot \mathcal{R}_{\text{torso\_height}}(\mathbf{s}) \cdot \mathcal{R}_{\text{upright}}(\mathbf{s}) \cdot \mathcal{R}_{\text{straight}}(\mathbf{s}) \cdot \mathcal{R}_{\text{control}}(\mathbf{a}) \tag{1}$$

where $\mathcal{R}_{\text{forward\_vel}}(\mathbf{s})$ encourages high forward velocity along the generated track, $\mathcal{R}_{\text{torso\_height}}(\mathbf{s})$ encourages the torso to maintain a height equivalent to an at-rest standing pose, $\mathcal{R}_{\text{upright}}(\mathbf{s})$ encourages the torso to maintain an upright pose, $\mathcal{R}_{\text{straight}}(\mathbf{s})$ encourages the agent to walk in the center of the track, and $\mathcal{R}_{\text{control}}(\mathbf{a})$ penalizes large actions (energy minimization). They find that optimizing a product of these five rewards is more effective than optimizing *e.g.* a sum of rewards.

## 4 Discussions

In this section, we briefly discuss the top two solutions presented in both tracks and provide insights on how their groundbreaking research informs human dexterity and agility movement and compensation for the loss of limbs.

**The manipulation track** with myoMPL poses several challenges for developing a control policy. First, the myoArm is high-dimensional and overactuated. Second, the start and goal conditions are randomized. Third, the handover process and hand-object interactions introduce additional constraints that must be managed for stable manipulation. Additionally, the environment provided by MyoChallenge 24' still poses several challenges to transitioning to real-world implications. Specifically, the environment assumes both the MPL and the myoArm have oracle information on each other's state, while in reality, that information is mainly communicated through vision to the user and through a myographic interface to the device.

Both of the top two teams chose to use muscle synergies to reduce the dimensionality of the action space and achieve more efficient and refined control of the finger movements. Those strategies, in comparison to the baseline solution trained based on curriculum learning, exhibit smoother motions during the reach and grasp of the myoArm. Muscle Heads utilized controlled variables to mimic the way precise pinches are performed, and the LNS Group leverages the DynSyn to accelerate training while maintaining interpretability. Moreover, both teams decided to separate the control of the MPL from the myoArm, using either inverse dynamics or trajectory interpolation. Although effective, this independent control poses a challenge to the handover component of the task as the two controllers are not communicating effectively when the object is released and grasped. As observed from the Muscle Heads team, the success rate of the final placement significantly decreases due to unsuccessful passovers. The final winning solution achieves only a score of $26\%$, highlighting the potential gap between existing models and true dexterous manipulation skills.

**The locomotion track** is also significantly more challenging than previous iterations of MyoChallenge. First, the terrain was highly varied, ranging from flat ground to stairs with step heights up to 24 cm. This diversity required a wide range of motor skills and introduced significant uncertainty as the agent sparsely observed the terrain with a heightmap. Second, the agent had no direct control over the prosthetic leg, which has an independent controller as a finite state machine. Lastly, the prosthetic leg's internal state was unobservable, with only the socket interaction force on the femur available. Additionally, although the environment and observables of myoOSL intend to be as realistic as possible, the socket interaction force might not accurately reflect real-world readings as MuJoCo simulates all components (e.g., human femur, prosthetic leg, and ground) as rigid bodies [46].

The winning team of the locomotion track uses imitation learning based on a large and diverse motion capture dataset to generate human-like movements, with most joint motions within physiological joint limits (Appendix A.7 - Fig. 8 A). This allows the myoOSL to match the biological joints of reference motion kinematically. However, the prosthetic leg's stance phase was notably shorter, visually deviating from typical human walking patterns (Appendix A.7 - Fig. 8 B). This observation indicates the need for further investigation into how prosthetic limbs influence walking style, effort, and stability. Future policies could be improved by leveraging internal prosthetic leg information (e.g., training a teacher network that observes the internal state and provides reference actions for the policy [47]). Additionally, both top-ranking teams utilize curriculum learning to improve generalization across varied terrains, highlighting the role of progressive training in mastering complex motion.

**Emphasis on physiological accuracy** While advances in machine learning have enabled greater agility and dexterity in bionic limbs, equal emphasis should be placed on ensuring physiological accuracy relative to experimental human data. This year's MyoChallenge openly encourages designing controllers that produce physiologically accurate motions. The manipulation track produced novel solutions inspired by muscle synergy principles [48], which did not rely solely on machine learning. In locomotion tasks, team MSKBioDyn's solution leverages imitation learning from real-world data to generate movements that significantly reduce overextension torque, without explicit torque penalties, effectively addressing physiological pain criteria. Although modeling pain as joint overextension within the locomotion track is a simplified proxy, it serves as a useful first-order approximation to stimulate discussion and development of physiologically plausible control strategies. More detailed metrics, such as stress distribution, swelling, or tissue damage [49, 50, 51], remain difficult to simulate, but the benchmark has already inspired follow-up studies on prosthesis interaction [52, 53]. For future developments, integrating more biologically realistic sensory feedback, such as muscle spindle models [54, 55], could further bridge the gap between human motor control and bionic limb performance.

## 5   Conclusion and Future Challenges

Digital twins of humans are indispensable tools for understanding neuromotor control, enabling cost-effective prototyping and controller design for bionic limbs. In this paper, we present MyoChallenge 2024: Physiological Dexterity and Agility in Bionic Humans—a competition aiming to benchmark dexterous manipulation and agile locomotion in prosthesis users. This iteration of myochallenge has successfully inspired several state-of-the-art controlling algorithms for both upper and lower limb control with external bionic limbs. The winning solutions show a variety of algorithms ranging from curriculum learning, imitation learning, model-based control, and biologically inspired muscle synergy. Nevertheless, these solutions still face challenges in generalizing to unseen environments

and achieving physiological fidelity for real-world limb loss users. In the upcoming myochallenge, we aim to target higher-order aspects of human motor control, including athletic intelligence. We also aim to broaden participation by lowering barriers for researchers from underrepresented groups and underdeveloped regions. We invite the global research community to join us in advancing neuromotor control and human-machine interaction by participating in future editions of this competition.

## Acknowledgments and Disclosure of Funding

We would like to acknowledge support for this competition from the University of Twente Techmed and DSI, Northeastern University - The Institute for Experiential Robotics (IER), Google Cloud Computing, Google-Deepmind, Össur, and EU ERC StG Interact. Special thanks goes to Dhruv Batra, Ram Ramrakhya, Deshraj Yadav, and Rishabh Jain for help with the EvalAI platform. A.C., A.M.V. and A.M.: Swiss SNF grant (310030_212516) G.D. and C.W.: NSERC Discovery Grant and NFRF: Exploration

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

# A  Appendix

## A.1  Competition Details

The competition ran from July 15th to November 21st, with a final workshop in the NeurIPS 2024 conference competition: MyoSymposium (`https://sites.google.com/view/myosuite/myosymposium/neurips24`). The workshop allowed winners from both tracks to present their solutions and bring together researchers and scholars in the field of biomechanics, ML, neuroscience, and health care.

## A.2  Musculoskeletal Models Details

The MPL system features 17 independent actuators controlling 26 degrees of freedom (DOFs) [25]. Each finger contains three coupled DOFs for flexion, with additional abduction DOFs in the index, ring, and little fingers (the latter two being mechanically coupled) [25]. The thumb's four DOFs comprise three flexion axes and one abduction axis, while the wrist provides three rotational DOFs [25]. The system includes single-DOF joints at both the elbow and humeral rotator, along with a two-joint shoulder complex enabling flexion-extension and abduction-adduction through serial hinge joints [25]. This configuration achieves near-human strength, speed, and kinematic fidelity [25].

The myoOSL model integrates a myoelectric residual limb with an OpenSim Leg (OSL) prosthesis [56, 57]. The OSL component precisely replicates its physical counterpart's 5.377 kg mass, peak joint torques (142.2 Nm knee, 168.2 Nm ankle), and functional ranges (0-120° knee flexion, ±30° ankle rotation) [56, 57]. The combined system's 23 DOFs include: 19 actively controlled DOFs (driven by 54 muscle actuators and two OSL torque actuators) and 4 passively constrained DOFs governing socket-residuum interaction [58].

## A.3  Environment Details

Table 2: Variations in Object Physical Properties for Manipulation Track. Default values follow that of the YCB object set [59]

| Property | Default Value | Variation |
|---|---|---|
| **Dimensions (mm)** | | |
| Width | 72 | $\pm$0–5% |
| Depth | 88 | $\pm$0–10% |
| Height | 28 | $\pm$0–5% |
| **Mass** | 97 | $\pm$50 grams |
| **Friction Coefficients** | | |
| Sliding | 1.0 | $\pm$0.1 |
| Torsional | 0.005 | $\pm$0.001 |
| Rolling | 0.0001 | $\pm$0.00002 |

**Manipulation Track Baseline** The baseline model is trained using Proximal Policy Optimization (PPO) [39] implemented in the `stable-baselines3` framework [60]. Training is performed on a single Intel i7 CPU core for 12 hours, employing a three-stage curriculum learning strategy to progressively increase task difficulty. We use the following hyperparameters: learning rate $\eta = 3 \times 10^{-4}$, update horizon $T = 2048$ steps, mini-batch size $B = 64$, discount factor $\gamma = 0.99$, PPO clip range $\epsilon = 0.2$, entropy coefficient of $0.03$ with Adam optimizer.

**Locomotion Track Baseline** No baseline is provided.

Table 3: Observation Space Dimensions for the myoMPL Manipulation Task

| Observation | Dimension |
|---|---|
| Elapsed time | 1 |
| Joint positions of myoArm | 38 |
| Joint velocities of myoArm | 38 |
| Joint positions of MPL | 26 |
| Joint velocities of MPL | 26 |
| Position and orientation of object | 7 |
| Velocities of object | 6 |
| Start position | 3 |
| Goal position | 3 |
| Object contact information | 5 |
| Muscle activations | 63 |
| Hand reaching error | 3 |
| Hand passing error | 3 |
| **Total** | **222** |

Table 4: Observation Space Dimensions for the myoOSL Locomotion Task

| Observation | Dimension |
|---|---|
| Elapsed time | 1 |
| Terrain type | 1 |
| Joint positions of myoLeg | 17 |
| Joint velocities of myoLeg | 17 |
| Ground Reaction forces of myoLeg | 2 |
| Socket forces | 3 |
| Pelvis angle (in world frame) | 4 |
| Muscle activations | 54 |
| Muscle lengths | 54 |
| Muscle velocities | 54 |
| Muscle forces | 54 |
| Planar Pelvis position (in world frame) | 2 |
| Planar Pelvis velocity (in world frame) | 2 |
| Terrain height map | 100 |
| **Total** | **365** |

## A.4 Computation Resources

Evaluation platform and support for the deployment of MyoChallenge are provided by eval.ai ([61]). Compute infrastructure to run the MyoChallenge are from Amazon Web Services (AWS), given in the list below.

- AWS EC2 - c5.4xlarge (16 CPU, 32 GB RAM)
- AWS EBS - gp2 (17 GB)
- AWS ECS for Kubernetes
- AWS ECR

## A.5   Manipulation First Place Detailed Solution

**Detailed Reward Design** The myo and robot hands were encouraged to grasp the object at different points on the object (top left and bottom right, respectively) to prevent their fingers from becoming entangled during handover. To release the object from the myoHand, rather than define a dedicated reward component, the muscle inputs were simply set to 0 after the robot hand had grasped the object.

More formally, in each stage $i = \{1, 2\}$ of the training curriculum, the total reward $r^{(i)} = \sum_j r_j^{(i)}$ was a sum of individual reward components $r_j^{(i)}$ (Table 5). The reward components were functions of the Euclidean distance between vectors $\boldsymbol{x}$ and $\boldsymbol{x}'$ (Table 6):

$$f(\boldsymbol{x}, \boldsymbol{x}', l) = \exp\{-l\|\boldsymbol{x} - \boldsymbol{x}'\|_2\}, \quad g(\boldsymbol{x}, \boldsymbol{x}') = \mathbb{1}(\|\boldsymbol{x} - \boldsymbol{x}'\|_2 < 0.005),$$
$$h(\boldsymbol{x}, \boldsymbol{x}', l) = f(\boldsymbol{x}, \boldsymbol{x}', l) + g(\boldsymbol{x}, \boldsymbol{x}'),$$

where $\mathbb{1}(\text{condition})$ is the indicator function that returns 1 if condition is true and 0 otherwise.

Table 5: Reward function components (Muscle Heads)

| | |
|---|---|
| $r_1^{(1)}$ | $h\big(\boldsymbol{x}_t^{\text{thumb (myo)}}, \boldsymbol{x}_t^{\text{object (top)}}, 5\big) \times \mathbb{1}(myoArm\ thumb\ behind\ object)$ |
| $r_2^{(1)}$ | $h\big(\boldsymbol{x}_t^{\text{index finger (myo)}}, \boldsymbol{x}_t^{\text{object (top)}}, 5\big) \times \mathbb{1}(myoArm\ index\ finger\ in\ front\ of\ object)$ |
| $r_3^{(1)}$ | $10 \times h\big([\boldsymbol{x}_t^{\text{thumb (robot)}} + \boldsymbol{x}_t^{\text{index finger (robot)}}]/2, \boldsymbol{x}_t^{\text{object (bottom)}}, 5\big) \times \mathbb{1}(object\ touching\ myoArm\ or\ robot)$ |
| $r_4^{(1)}$ | $0.25 \times f\big(\boldsymbol{\theta}_t^{\text{object}}, \boldsymbol{\theta}_0^{\text{object}}, 0.5\big)$ |
| $r_1^{(2)}$ | $h\big(\boldsymbol{x}_t^{\text{thumb (robot)}}, \boldsymbol{x}_t^{\text{object (bottom)}}, 5\big) + h\big(\boldsymbol{x}_t^{\text{index finger (robot)}}, \boldsymbol{x}_t^{\text{object (bottom)}}, 5\big)$ |
| $r_2^{(2)}$ | $10 \times \mathbb{1}(object\ touching\ robot) \times \mathbb{1}(robot\ thumb\ \&\ index\ finger\ on\ opposite\ sides\ of\ object)$ |

Table 6: Reward function variables (Muscle Heads)

| | |
|---|---|
| $\boldsymbol{x}_t^{\text{thumb (myo)}}$ | position of the myoHand thumb tip |
| $\boldsymbol{x}_t^{\text{index finger (myo)}}$ | position of the myoHand index finger tip |
| $\boldsymbol{x}_t^{\text{thumb (robot)}}$ | position of the robot thumb tip |
| $\boldsymbol{x}_t^{\text{index finger (robot)}}$ | position of the robot index finger tip |
| $\boldsymbol{x}_t^{\text{object (top)}}$ | position of top left corner of object |
| $\boldsymbol{x}_t^{\text{object (bottom)}}$ | position of bottom right corner of object |
| $\boldsymbol{\theta}_t^{\text{object}}$ | orientation of object (Euler angles) |

**Computation Resources** Training was performed on a single GPU with 20GB RAM. 20 CPU cores were used to simulate environments in parallel. Training took approximately 14 hours to perform 7M steps.

**Training and Test Details** The policy used to control the myoArm and robot hand was trained using the SBX (Stable Baselines Jax) implementation of Soft Actor Critic (SAC)[62] with the Adam optimizer. The following hyperparameters were used: learning rate $1 \times 10^{-4}$, number of hidden units in both the policy and the value feedforward networks [256, 256], discount factor 0.99, soft update coefficient 0.02, buffer size $2.5 \times 10^5$, batch size 256, train frequency 25 and gradient steps -1. In total, the policy was trained for 7M steps (agent-environment interactions). Stage 1 of learning lasted for 3.5M steps, and stage 2 of learning lasted for another 3.5M steps. In stage 3, the robot hand was moved to the goal location using inverse kinematics. This was achieved by performing gradient descent on the distance between the desired and actual position of the robot hand with respect to the joint angles of the robot arm. To avoid pillar collisions, the hand was moved to the goal location via a waypoint located above the pillar.

## A.6 Manipulation Second Place Detailed Solution

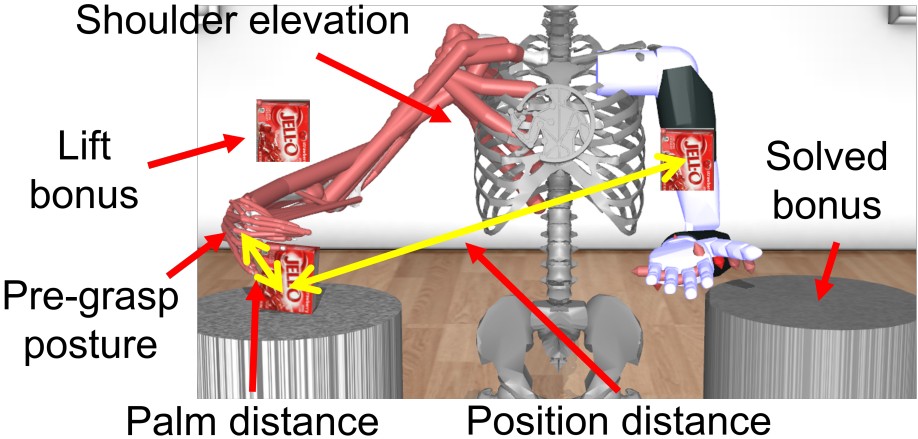

Figure 7: The demonstration of reward terms.

**Detailed Weight Design** While the actuators within a group perform shared actions, state-dependent correction weights are introduced to allow fine-tuned adjustments for individual actuators. Specifically, a unified action is generated for each group, along with state-dependent correction weights for each actuator. The final action is computed by combining the unified action with the correction weights, ensuring both efficient exploration and precise control. Additionally, the state-dependent adaptation allows the agent to perform task-specific adjustments, improving motor control in complex environments.

**Detailed Reward Design**

- Position distance: This reward term measures the Euclidean distance between the object and the target position. We encourage the agent to minimize this distance.

- Palm distance: This term penalizes the distance between the palm and the object to promote stable grasping.

- Pre-grasp posture: Encourage the hand to achieve a pre-grasp posture, defined as the alignment of the palm and fingers relative to the object.

- Lift bonus: Binary reward is granted when the object is lifted above a predefined threshold.

- Shoulder elevation: Penalizes deviation from initial shoulder elevation angle, maintaining stable arm posture.

- Solved bonus: A sparse reward granted upon task completion (object reaches the goal position).

The structured reward function effectively guided the agent in balancing task objectives and stability constraints. The agent successfully completed the object transfer task within 10 million training steps, achieving fast convergence without the need for curriculum learning. The reduced training time also facilitated rapid tuning of reward weights, allowing for quick adaptation to different task conditions.

**Computation Resources** The training was conducted on an NVIDIA A100 GPU paired with an Intel(R) Xeon(R) Gold 6348 CPU, utilizing 64 parallel environments. A total of 10 million training steps were completed in approximately 3.5 hours.

**Training and Test Details** The code and configuration for training and evaluation can be found at https://github.com/zchJo/MyoChallenge-2024-DynSyn.

## A.7  Locomotion First Place Detailed Solution

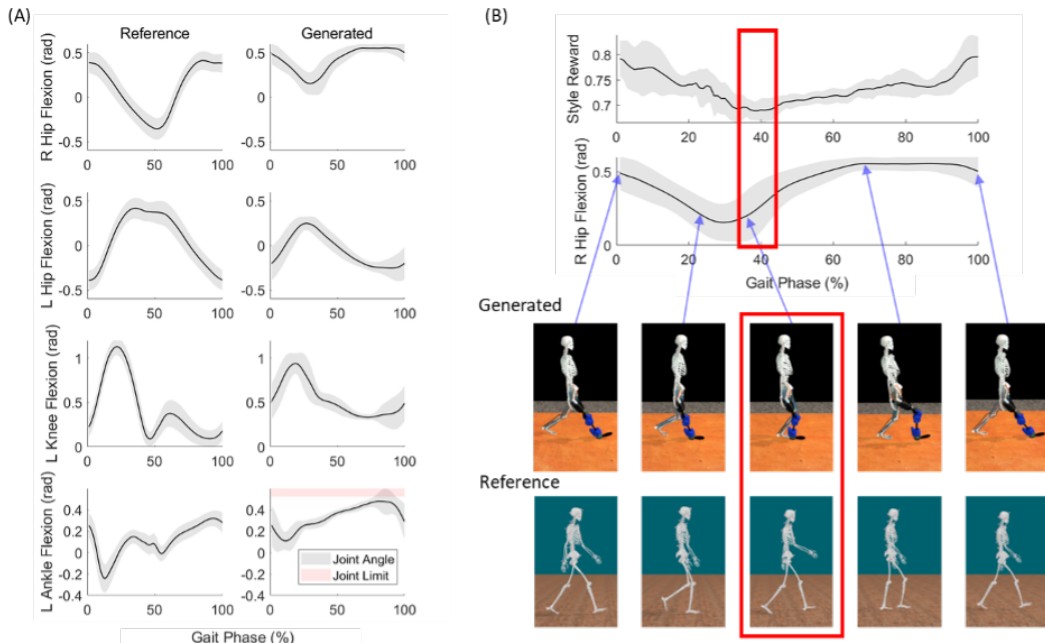

Figure 8: Motion generated by team MSKBioDyn. (A) Kinematics of four biological joints (right hip flexion, left hip flexion, left knee flexion, and left ankle flexion) compared to the reference training data. Joint limits (red areas) are omitted if the joint angles maintain a sufficient margin (> 0.2 rad) from the limits. (B) Visual comparison of the walking motion with the reference. The left heel strike phase (highlighted by a red box) notably deviates from the reference, resulting in lower style rewards.

**Computational Resources** Team MSKBioDyn trained their solution on a single Ubuntu server equipped with two AMD EPYC 9654 CPUs (96 cores each), a GeForce RTX 4090 GPU with 24GB of memory, and 384GB of DDR5 RAM. The dynamics computations, performed using MyoSuite and MuJoCo, were executed on the CPU, and all neural network components, including the PPO algorithm, were implemented in PyTorch with CUDA acceleration on the GPU. Although the machine had 384GB of system memory, the training script utilized less than 50GB, making the results reproducible on systems with smaller memory capacity.

**Training and Test Details** The policy was trained using a simulation environment built on the MuJoCo C++ library (version 3.2.2), incorporating the human musculoskeletal model provided by MyoSuite. A custom Python script handled both environment parallelization and neural network training. The training was performed using 500 parallel environments, with episode durations ranging from 2.0 to 8.0 seconds depending on the stage of the curriculum. The curriculum was manually scheduled based on task performance, such as survival time and average rewards. All neural networks were trained using the Adam optimizer, and adaptive learning rate scheduling was applied for the actor and critic networks. Further hyperparameter settings for PPO and AMP are summarized in Table 7.

**Dataset and Github Link** The solution, including trained policy and terrain encoder networks, is available at the following link `https://github.com/gparc/myochallenge_2024eval_msk`. The motion capture dataset used in imitation learning is also publicly available [63]. The training environment, which comprises the implementation of the PPO and the imitation learning framework, remains confidential.

Table 7: Hyperparameter Settings for AMP and PPO

| Parameter | Value |
|---|---|
| AMP Task reward weight | 0.1-0.4 |
| AMP Style reward weight | 0.6-0.9 |
| AMP Gradient penalty | 10.0 |
| PPO Samples per update | 200k-800k |
| PPO Batch size | 4096 |
| AMP Discriminator batch size | 2048 |
| PPO Learning rate | 1e-5 to 1e-4 |
| AMP Discriminator learning rate | 1e-5 |
| PPO Discount factor | 0.998 |
| PPO Clipping range | 0.2 |
| PPO Value loss coefficient | 1.0 |
| PPO Entropy coefficient | 0.0 |
| PPO Maximum gradient norm | 0.5 |

## A.8 Locomotion Second Place Detailed Solution

**TD-MPC2 World Model** The world model consists of five components:

$$
\begin{aligned}
\text{Encoder} \quad & \mathbf{z} = h(\mathbf{s}) && \triangleright \text{ Encodes state into a latent embedding} \\
\text{Latent dynamics} \quad & \mathbf{z}' = d(\mathbf{z}, \mathbf{a}) && \triangleright \text{ Predicts next latent state} \\
\text{Reward} \quad & \hat{r} = R(\mathbf{z}, \mathbf{a}) && \triangleright \text{ Predicts reward } r \text{ of a state transition} \\
\text{Terminal value} \quad & \hat{q} = Q(\mathbf{z}, \mathbf{a}) && \triangleright \text{ Predicts discounted sum of rewards} \\
\text{Policy prior} \quad & \hat{\mathbf{a}} = p(\mathbf{z}) && \triangleright \text{ Predicts an action } \mathbf{a}^* \text{ that maximizes } Q
\end{aligned}
\tag{2}
$$

where $\mathbf{z}$ is a latent state. Components of the world model are trained end-to-end using interaction data collected in an online RL manner. The official and publicly available implementation at https://www.tdmpc2.com/ is used.

**Training Curriculum** Team Loco UCSD uses a simple training curriculum by first learning to walk on flat terrain, and then subsequently starting to randomize the terrain by varying the slope of consecutive segments. They empirically find that this leads to faster convergence and thus lower training wall-time.

**Computational Resources** Training the locomotion policy takes approximately 2 days on a single NVIDIA RTX 3090 GPU using 4 parallel environments. CPU and RAM usage is neglible. Terrain randomization was enabled after 1 day of training, i.e., half-way through. They did not find it necessary to experiment with other infrastructure, configurations, nor hyper-parameters.

**Hyperparameters** Team Loco UCSD uses default hyperparameters wherever applicable. However, key hyperparameters are listed in Table 8 for completeness.

Table 8: Hyperparameters for Team Loco UCSD's solution using TD-MPC2.

| Parameter | Value |
|---|---|
| Parameters | 5M |
| Architecture | MLPs |
| Activation | LayerNorm+Mish |
| Latent space | SimNorm |
| Batch size | 256 |
| Learning rate | 3e-4 |
| Discount factor | 0.99 |
| Parallel envs | 4 |
| Environment steps | 10M |
| Buffer size | 1M |

## A.9 Tutorials and Baseline

Throughout the competition, we provided several colab tutorials, video instructions, baseline as well as workshops, as summarized below.

**Colab Tutorials:**

- MyoChallenge Tutorial1 - Getting Started with MyoSuite: `https://colab.research.google.com/drive/1AqC1Y7NkRnb2R1MgjT3n4u02EmSPem88?usp=sharing`
- MyoChallenge Tutorial2 - Getting Started with Baselines: `https://colab.research.google.com/drive/1YJqhKWKNJ6MFUKqTQYLilc9M6BdBfA3g?usp=sharing`
- MyoChallenge Tutorial3 - Submission Instructions: `https://colab.research.google.com/drive/11vRvWMWykNrd_5ViJVGdLXz2pnbc5QEs?usp=sharing`
- MyoChallenge Tutorial4 - Loading Latest Baseline: `https://colab.research.google.com/drive/1vHp7aK8vkhWOwknf-VHeENquHfb86qOP?usp=sharing`

**Manipulation Baselines:**

- Baseline Download URL: `https://drive.google.com/drive/folders/1c1pTdH10LfGdz9Wrb-P3o4iHWqGXtpZZ?usp=sharing`
- Baseline rendering: `https://youtu.be/332TcmMUABA?si=Zeag_wnrqRFxxmZ`

**MyoChallenge 24' Workshop and Q&A:**

- MyoChallenge 24' Workshop and Q&A: `https://youtu.be/bjZomRsV5Ac?si=Uwjzu3Cj8C4ug4BJ`
- MyoSuite/MyoChallenge: Towards Human Embodied Intelligence: `https://youtu.be/uQ2QZznae8M?si=0hBdfroqnKwDVfCa`
- MyoChallenge Q&A and Announcement Thread: `https://github.com/MyoHub/myosuite/discussions/206`

**Documentation:**

- Official MyoChallenge '24 Documentation: `https://myosuite.readthedocs.io/en/latest/challenge-doc.html`
- An all-in-one comprehensive guide by Tatsuki Tsujimoto: `https://ttktjmt.com/blog/comprehensive-guide-to-myochallenge-2024/`

