# OpenReview forum: "MyoChallenge 2024: A New Benchmark for Physiological Dexterity and Agility in Bionic Humans"
_NeurIPS.cc/2025/Datasets_and_Benchmarks_Track — NeurIPS 2025 Datasets and Benchmarks Track poster_

### Official Review · Reviewer_jMG3 · 2025-06-09

**Rating:** 5
**Confidence:** 3

**Summary:**

This paper presents MyoChallenge 2024, a well-know benchmark competition at NeurIPS focusing on bionic human motor control. The competition includes two tracks: 1) manipulation (coordinating a biological arm with a Modular Prosthetic Limb) and 2) locomotion (navigating terrains with a transfemoral amputation model and an Open Source Leg).
I think MyoChallenge 2024 is a valuable bionic motor control benchmarks. The open-source resources and competition outcomes serve as foundations for future research in for bionic limb control and assistive technologies.

**Dataset Code Accessibility:**

Yes

**Dataset Code Comments:**

The competition website is provided at the end of the Abstract, with the accessible dataset provided.

**Ethical Comments:**

There are no significant ethical concerns, as this competition focuses on simulated environments and digital models, with no involvement of human participants, patients, or clinical trials.

**Ethical Considerations:**

No, there are no or only very minor ethics concerns

**Final Justification:**

The rebuttal addresses most of my concerns. I’ll keep my opinion as accept.

**Limitations Weaknesses:**

I think there are 3 main strengths/contributions in this paper:


### 1. **Overly Idealized Simulation Assumptions**
The simulation environment assumes **perfect, noiseless state synchronization** between biological and prosthetic components (e.g., instant knowledge of joint states and object interactions), which contrasts with real-world scenarios where **sensory feedback (e.g., myoelectric signals, visual perception)** is noisy, delayed, and incomplete. The absence of sensor noise or latency in the model may **overstate algorithm robustness** and underestimate challenges in translating simulated solutions to physical prosthetics.


### 2. **Limited Participant Success Over Baselines**
Only **5.7% (3/53 teams)** surpassed the manipulation baseline, and **3.7% (2/53 teams)** exceeded the locomotion baseline, indicating that task complexity may be misaligned with the broader participants‘ capabilities.

**Strengths Contributions:**

I think there are 3 main strengths/contributions in this paper:

### 1. **Task Design Aligned with Prosthetic Functionality**
The competition introduces two core tasks—**manipulation** (bimanual object handover with MPL) and **locomotion** (terrain navigation with transfemoral amputation)—mimicking real-world prosthetic use cases to address dexterity and mobility challenges.


### 2. **High-Fidelity Simulation Reducing Realism Gap**
Using the MyoSuite framework (MuJoCo), the benchmark employs **physiologically realistic musculoskeletal models** (e.g., myoArm with 63 muscle units) and dynamic environments (randomized terrain/object properties), enabling scalable, biologically accurate simulations to bridge simulation and real-world performance.


### 3. **Accessible Baselines and Open Ecosystem**
The competition provides **standardized baselines** (e.g., RL-based manipulation policy), open-source tools (MyoSuite, GitHub templates), and diversity-focused awards (DEI/Student Awards), fostering inclusivity and community-driven innovation in bionic limb research.

---

> ### Author Rebuttal · Authors · 2025-07-31
>
> We thank the reviewer for the feedback, and we appreciate the recognition of MyoChallenge 2024 as a valuable benchmark.
>
> > 1. Overly Idealized Simulation Assumptions
> > The simulation environment assumes perfect, noiseless state synchronization between biological and prosthetic components (e.g., instant knowledge of joint states and object interactions), which contrasts with real-world scenarios where sensory feedback (e.g., myoelectric signals, visual perception) is noisy, delayed, and incomplete. The absence of sensor noise or latency in the model may overstate algorithm robustness and underestimate challenges in translating simulated solutions to physical prosthetics.
>
> We appreciate the reviewer’s observation regarding the idealized sensory assumptions in the current simulation. Our current design choice of a synchronized, full-state observation intends to create a benchmark to first tackle the challenge of musculoskeletal-prosthetic coordination, a problem that is already non-trivial due to the system’s dimensionality, contact dynamics, and actuator complexity. Introducing noisy or partial observation at this stage could risk additional complexity that the broader community could yet handle.
>
> Nevertheless, we fully agree that achieving robust control under partial and noisy observations is a critical next step. As current success rates remain low, future editions of MyoChallenge will incorporate more realistic sensor models, particularly as performance improves, to enhance diversity and better reflect real-world constraints (as also discussed in Section 4 of the submission).
>
>
> > 2. Limited Participant Success Over Baselines
> > Only 5.7% (3/53 teams) surpassed the manipulation baseline, and 3.7% (2/53 teams) exceeded the locomotion baseline, indicating that task complexity may be misaligned with the broader participants‘ capabilities.
>
> We acknowledge that both tracks in this edition of MyoChallenge were highly challenging by design to push the boundaries of human-computer interaction proposed for NeurIPS 2024. Akin to all previous grand-challenges, the current success rate demonstrates that its instantiation captures these inherent difficulties of the task, solving which will indicate a large step forward. The current winning solution, while capable, also indicates that there is a large room for improvement, and the challenge will remain relevant for the upcoming developments.
>
> Additionally, we plan to introduce a two-phase approach for the MyoChallenge of the NeurIPS 2025 edition, in which the first phase introduces minimum variations and allows low barriers of entry and a wider range of participation, and the second phase with added difficulties to allow for the development of more robust algorithms for more advanced teams.

---

> > ### Comment · Reviewer_jMG3 · 2025-08-02
> >
> > Thanks for your rebuttal and it addresses most of my concerns. I think it’s clear that MyoChallenge 2024 is a valuable benchmark for the community, even under the idealized sensory assumptions. I’ll keep my opinion as accept.

---

### Official Review · Reviewer_DGbh · 2025-06-21

**Rating:** 4
**Confidence:** 3

**Summary:**

The paper presents MyoChallenge 2024, a competition and benchmark designed to advance research in controlling bionic limbs integrated with the human body. The authors introduce two novel, physiologically-realistic simulation tracks within the MyoSuite framework: a manipulation track featuring a musculoskeletal arm and a prosthetic arm performing a handover task, and a locomotion track where a transfemoral amputee model walks on varied terrain. The paper details the design of these benchmarks and analyzes the state-of-the-art solutions developed by the winning teams.

**Dataset Code Accessibility:**

Yes

**Ethical Considerations:**

No, there are no or only very minor ethics concerns

**Final Justification:**

In light of the rebuttal and the general consensus among the reviewers, I have raised the score to 4.

**Limitations Weaknesses:**

1. The authors emphasize physiological realism, yet the metrics used are highly simplified proxies for complex biological phenomena. For instance, "pain" is modeled solely as joint overextension torque, which largely ignores the primary source of discomfort for amputees: the physical interaction at the socket-residuum interface. The authors rightly note that MuJoCo's rigid-body simulation cannot fully capture these crucial forces. Relying on these simplified metrics risks promoting solutions that are optimal in simulation but physiologically irrelevant or even detrimental in the real world, thus weakening the benchmark's claimed translational value.
2. The design of the manipulation track appears to incentivize highly engineered, multi-stage solutions rather than robust, generalizable policies. The winning teams employed task-specific modules like inverse kinematics and predefined trajectories to solve distinct phases of the task. The fact that the top-performing solution achieved only a 26% success rate suggests these engineered approaches are brittle. The current setup may not effectively distinguish between a truly adaptive policy and a carefully scripted one.
3. The explicit decision not to provide a baseline controller for the highly complex locomotion track is a significant weakness. This creates a formidable barrier to entry for research groups, particularly those without access to large-scale, private motion-capture datasets like the one used by the winning team. The absence of a common starting point makes it difficult to fairly compare the performance of different algorithmic approaches. To foster broader participation and ensure a level playing field for future research, it is strongly recommended that the authors provide at least a simple, open-source baseline agent, even if its performance is modest.

**Strengths Contributions:**

1. The authors tackle a well-motivated and important problem by creating a benchmark for the integrated control of both upper- and lower-limb prosthetics with complex musculoskeletal models, a domain that prior benchmarks have largely overlooked.
2. The competition introduces novel high-fidelity models (myoMPL, myoOSL) and contact-rich tasks (bimanual handover, varied terrain) that are demonstrably challenging and effectively test the limits of current control algorithms.
3. The paper is well-written, organized, and easy to understand.

---

> ### Author Rebuttal · Authors · 2025-07-31
>
> We thank the reviewer for their thoughtful suggestions and insights.
>
> > The authors emphasize physiological realism, yet the metrics used are highly simplified proxies for complex biological phenomena. For instance, "pain" is modeled solely as joint overextension torque, which largely ignores the primary source of discomfort for amputees: the physical interaction at the socket-residuum interface. The authors rightly note that MuJoCo's rigid-body simulation cannot fully capture these crucial forces. Relying on these simplified metrics risks promoting solutions that are optimal in simulation but physiologically irrelevant or even detrimental in the real world, thus weakening the benchmark's claimed translational value.
>
> We thank the reviewer for this thoughtful observation and for noting that we have already discussed these limitations in the submission. We will further clarify the reviewer’s concern here.
>
> We fully agree that modeling pain purely as joint overextension is a simplified approximation and does not capture the full complexity of discomfort experienced by amputees. We are aware of promising work, such as [Gupta2020], [McGeehan2023], and [Matray2025], which explore more detailed quantification of human-prosthesis interaction using metrics like stress distribution, swelling, and tissue damage. Nevertheless, given the limited progress in simulating such pain, we adopted this proxy as a first-order approximation that is meant to encourage the development of physiologically plausible control strategies and stimulate discussion around safety and comfort in assistive technologies.
>
> We hope such developments will inform future iterations of MyoChallenge. Encouragingly, the benchmark has already spurred meaningful follow-up work in the community for prostheses interactions, including efforts such as [Zou2023] and [Tan2025].
>
> > The design of the manipulation track appears to incentivize highly engineered, multi-stage solutions rather than robust, generalizable policies. The winning teams employed task-specific modules like inverse kinematics and predefined trajectories to solve distinct phases of the task. The fact that the top-performing solution achieved only a 26% success rate suggests these engineered approaches are brittle. The current setup may not effectively distinguish between a truly adaptive policy and a carefully scripted one.
>
> We appreciate the reviewer’s concern regarding the reliance on engineered, multi-stage solutions. However, we would like to emphasize that the design of the manipulation track is agnostic to the solution strategy. Participants were free to employ any approach, whether highly structured or fully learned, to tackle the task. The benchmark does not favor scripted methods, but instead sets up a complex environment that exposes the strengths and limitations of different control paradigms.
>
> We view the presence of hand-engineered components in early solutions as a natural progression in challenge-driven research. For example, the early stages of the ImageNet challenge saw dominance by hand-crafted features, which were ultimately surpassed by generalizable deep learning models. Similarly, we expect more robust and adaptive methods to emerge as the community continues to engage with this benchmark. To support this progression, we will open-source the benchmark and evaluation suite, making it fully accessible for future research and development in this space.
>
> Although the top-performing solution on the manipulation track used classical methods such as inverse kinematics, these methods were only used for a small fraction of the task, namely, repositioning the robot hand and releasing the object from the robot hand onto the table. The most challenging aspects of the task, controlling the 63-muscle myoArm and grasping the object with the robot fingers, still involved learning-based approaches.
>
> Additionally, to disincentivize highly engineered solutions, many aspects of the task were randomized across episodes, including the start and goal object/pillar locations, as well as the (unobserved) properties of the object, such as weight, size, and friction. The winning team noted that a substantial number of failed episodes were caused by the inverse kinematics failing to place the object stably on the randomized goal pillar. It is likely that a fully learned policy would not fall prey to these issues. Therefore, we believe that the current setup can effectively distinguish between a truly adaptive policy and a carefully scripted one. Moreover, hand-crafted and learned policies could be further distinguished by including different objects that require different grasps, which our setup permits and we have explored elsewhere [CaggianoA2023].
>
>
> > The explicit decision not to provide a baseline controller for the highly complex locomotion track is a significant weakness. This creates a formidable barrier to entry for research groups, particularly those without access to large-scale, private motion-capture datasets like the one used by the winning team. The absence of a common starting point makes it difficult to fairly compare the performance of different algorithmic approaches. To foster broader participation and ensure a level playing field for future research, it is strongly recommended that the authors provide at least a simple, open-source baseline agent, even if its performance is modest.
>
> We appreciate the reviewer’s concern and agree that providing a baseline controller is valuable for lowering the barrier to entry and supporting fair comparison across approaches. While we did not include an explicit baseline for this edition of the locomotion track in the manuscript, we did make available to all participants the full lower-limb model and baseline controller from the previous challenge (MyoChallenge 2023 – Chase Tag), implemented using the deprl framework [Schumacher2022]. Feedback from early adopters has confirmed that this prior baseline served as a helpful starting point.
>
> While the winning solution in locomotion used a private motion-capture dataset, using a public motion-capture dataset for imitation learning is also feasible. For example, the CMU Motion Capture Database [CMUMoCap] contains a set of motion sequences suitable for training a controller for the locomotion task. It includes level walking by diverse subjects, as well as slope walking (e.g., subject #74) and stair walking (e.g., subject #83). Although the large-scale dataset provided an advantage during training, we believe that the CMU Motion Capture Database is sufficient for generating walking motions through imitation learning [CaggianoB2023]. More importantly, the motion-capture dataset used by the winning team has now been released and is publicly available [Boo2025]. We will revise the manuscript to include the dataset availability in Appendix A.7, along with clearer documentation of the baseline resources available to participants.
>
> For future iterations, we plan to include an official markerless motion tracking software that allows participants to create their own database based on their solutions.
>
>
>
> Citations:
>
> [Zuo 2023] Zuo, C., He, K., Shao, J., & Sui, Y. (2023). Self model for embodied intelligence: Modeling full-body human musculoskeletal system and locomotion control with hierarchical low-dimensional representation.
>
> [Tan2025] Tan, C. K., Wang, C., Lyu, S., Hodossy, B. K., Schumacher, P., Wilson, E. B., … Song, S. (2025, May 12). Myoassist 0.1: Myosuite for dexterity and agility in bionic humans. 2025 International Conference On Rehabilitation Robotics (ICORR), 437–442.
>
> [Gupta2020] Gupta, S., Loh, K. J., & Pedtke, A. (2020). Sensing and actuation technologies for smart socket prostheses. Biomedical Engineering Letters, 10(1), 103–118.
>
> [McGeehan2023] McGeehan, M. A., Adamczyk, P. G., Nichols, K. M., & Hahn, M. E. (2022). A simulation-based analysis of the effects of variable prosthesis stiffness on interface dynamics between the prosthetic socket and residual limb. Journal of Rehabilitation and Assistive Technologies Engineering
>
> [Matray2025] Matray, M., Bonnet, X., Rohan, P.-Y., Calistri, L., & Pillet, H. (2025). Evaluating interface pressure in a lower-limb prosthetic socket: Comparison of FEM and experimental measurements on a roll-over simulator. Journal of Biomechanics
>
> [CaggianoA2023] Vittorio Caggiano, Sudeep Dasari, and Vikash Kumar. 2023. MyoDex: a generalizable prior for dexterous manipulation. ICML'23, Vol. 202. JMLR.org, Article 135, 3327–3346.
>
> [Schumacher2022] Schumacher, P., Häufle, D., Büchler, D., Schmitt, S., & Martius, G. (2022). DEP-RL: Embodied exploration for reinforcement learning in overactuated and musculoskeletal systems.
>
> [CMUMoCap] mocap.cs.cmu.edu.
>
> [CaggianoB2023] Caggiano, V., Durandau, G., Wang, H., Tan, C. K., Schumacher, P., Wang, H., ... & Kumar, V. (2024). Myochallenge 2023: Towards human-level dexterity and agility.
>
> [Boo2025] Boo, J., Seo, D., Kim, M., & Koo, S. (2025). Comprehensive human locomotion and electromyography dataset: Gait120. Scientific Data, 12(1), 1023.

---

> > ### Comment · Reviewer_DGbh · 2025-08-01
> >
> > Thank you for your rebuttal, which provided helpful context for some of the key design decisions I had questioned. In light of this information and the general consensus among the reviewers, I am updating my recommendation for this paper.

---

### Official Review · Reviewer_vcQq · 2025-07-02

**Rating:** 5
**Confidence:** 4

**Summary:**

This paper introduces a new open-source simulation benchmark for synergic control of human limb and prosthesis. This benchmark includes two tracks, manipulation and locomotion. The human limb was based on a high-fidelity skeleton-muscle model.

**Dataset Code Accessibility:**

Yes

**Ethical Considerations:**

No, there are no or only very minor ethics concerns

**Final Justification:**

Thank authors for their responses. I don't have any other questions and will keep my score.

**Limitations Weaknesses:**

* This benchmark is only able to simulate the biomechanics and cannot simulate the human neural control system, which, however, is indeed too difficult to realize and is out of the scope of this benchmark.
* It is uncertain that if the control algorithm trained in the simulation can be directly applied to the real prosthesis.
* The task may be too difficult and only two or three teams could beat the baseline. It may be better to design curriculum tasks. Take the locomotion for an example: The task can start from balance control on standing, then walking control on level ground, and finally the locomotion control on complex terrains.

**Strengths Contributions:**

* This benchmark addressed a major bottleneck in prosthetic research: the difficulty and risk of large scale experiments with real amputee subjects.
* This benchmark allows to train control models to finish tasks and minimize the muscle cost, which may provide insights to the realistic control to minimize the metabolic cost.
* The high-fidelity musculoskeletal models reduces the gap between the simulation and the biomechanics of real amputees.

---

> ### Author Rebuttal · Authors · 2025-07-30
>
> We thank the reviewer for the feedback and the thoughtful observations. Below, we provide some clarifications and context for each point:
>
> > This benchmark is only able to simulate the biomechanics and cannot simulate the human neural control system, which, however, is indeed too difficult to realize and is out of the scope of this benchmark.
>
> We agree with the reviewer that simulating a human neural control system would be a valuable long-term goal. However, our current benchmark intentionally focuses on biomechanical simulation as a practical and necessary first step. This is because the grand goal of this challenge is to design controllers for exoskeleton applications, where only external signals such as joint angles and forces are accessible. In addition, evaluation of the controller's performance can be done at the physics level, even without the use of a neural layer. While incorporating neural control would certainly deepen insights, our current benchmark provides a strong foothold for future integration as advancements in that area continue to emerge.
>
> > It is uncertain that if the control algorithm trained in the simulation can be directly applied to the real prosthesis.
>
> We recognize the reviewer’s concern and would like to point out that Myochallenge does not claim immediate transferability of learned policies to real prosthesis but instead aims to minimize the sim-to-real gap by emphasizing physiological realism. Although previous sim2real efforts in robotics have been successful, prosthesis devices are different in their physical interfaces and integration with human users. Hence, we design the challenge to focus on the deficiency in this subfield, and we hope to highlight the cap and draw more users to advance prosthetic control and assistive technology.
>
> > The task may be too difficult and only two or three teams could beat the baseline. It may be better to design curriculum tasks. Take the locomotion for an example: The task can start from balance control on standing, then walking control on level ground, and finally the locomotion control on complex terrains.
>
> We acknowledge that both tracks in this edition of MyoChallenge were highly challenging by design to push the boundaries of human-computer interaction here at NeurIPS 2025. That being said, we agree with the reviewer’s excellent suggestion. In fact, many top teams internally used curriculum learning to decompose the task into subtasks (e.g., reach, grasp, handover for manipulation in Section 3.1.1) and progressively scale difficulty of terrain (e.g., locomotion in 3.2.1). For future editions of MyoChallenge, we plan to incorporate two phases, in which the first phase introduces minimum variations and allows low barriers of entry, and the second phase, with added difficulties to allow for the development of more robust algorithms.

---

### Official Review · Reviewer_ZrjQ · 2025-07-02

**Rating:** 5
**Confidence:** 3

**Summary:**

The authors established MyoChallenge 2024 as a benchmark for human-robot coordination. Specifically, the benchmark focused on coordinating the biological limbs with the mechanical limbs under both manipulation and locomotion scenarios. The challenge attracted a wide range of participants and managed to significantly improve the coordination performance. Along with the challenge, a well-developed open-source simulation framework and standardized tasks were introduced, facilitating the development of the research community.

**Dataset Code Accessibility:**

Yes

**Ethical Considerations:**

No, there are no or only very minor ethics concerns

**Final Justification:**

The authors' rebuttal has addressed my concerns. I would like to keep my rating.

**Limitations Weaknesses:**

- Despite the progress, the final performance is still far from satisfactory. Moreover, the gap between simulation and real-world application is also noticeable. Further explorations based on these experiences would be promising.

- Decomposing the locomotion tasks and manipulation tasks is a practical divide-and-conquer strategy in reducing the overall complexity. However, a whole-body holistic physiological simulation, related infrastructures, and tasks are still expected to explore the upper-lower body coordination and whole-body manipulation.

**Strengths Contributions:**

- The open-sourced simulation framework MyoSuite is meaningful as a bridge between ML and biomechanics. Its efficiency, compatibility, and physiological fidelity are appreciated in providing a good entry point for the ML community to biomechanics and expanding the community to a broader domain.

- The designed tasks, with their focus on human-robot coordination, are challenging and meaningful in advancing the exploration of assistive robots and healthcare.

- The solutions demonstrate diverse technical routes, including biologically inspired synergy, curriculum learning, imitation learning, and world model, inspiring future progress.

---

> ### Author Rebuttal · Authors · 2025-07-30
>
> We thank the reviewer for the feedback, and we appreciate the recognition of MyoChallenge 2024 as a meaningful benchmark and a bridge between machine learning and biomechanics.
>
> > Despite the progress, the final performance is still far from satisfactory. Moreover, the gap between simulation and real-world application is also noticeable. Further explorations based on these experiences would be promising.
>
> We thank the reviewer for acknowledging that this is a relevant problem. Akin to all previous grand-challenges, the current success rate demonstrates that its instantiation captures these inherent difficulties of the task, solving which will indicate a large step forward.
>
> The current winning solutions, while capable, also indicate that there is a large room for improvement, and the challenge will remain relevant for the upcoming developments. Although recent advancements in sim2real transfer have shown success in relatively simple pick-and-place tasks, achieving sim-to-real transfer for full-body control, particularly in complex manipulation, remains a major challenge. We hope that this challenge can catalyze the ongoing progress in the field in two key ways: (1) provide a sim2real pipeline in complex real-world settings to compensate for the high cost of human experiments in real-world scenarios, (2) draw attention to the potential impact this MSK space can create.
>
> We also note that several research groups have been working in this direction based on the outcome of this challenge, such as [He2024] and [Tan2025].
>
> > Decomposing the locomotion tasks and manipulation tasks is a practical divide-and-conquer strategy in reducing the overall complexity. However, a whole-body holistic physiological simulation, related infrastructures, and tasks are still expected to explore the upper-lower body coordination and whole-body manipulation.
>
> We appreciate the reviewer for pointing this out and fully share the reviewer’s vision. In this edition of MyoChallenge, we intentionally separated manipulation and location to lower the barrier to entry, especially given the extra difficulty added on by prosthesis interactions. However, we would like to point out that our infrastructure, built on MyoSuite, already supports whole-body musculoskeletal models and has been tested on full-body tasks such as balancing and walking. In addition, in this year's (2025) edition of MyoChallenge, we have introduced tasks involving the movement of whole body coordination to achieve sports-level agility. We hope this helps clarify the reviewer’s concern.
>
>
> Citations:
>
> [He2024] He, K., Zuo, C., Ma, C., & Sui, Y. (2024). DynSyn: Dynamical Synergistic Representation for efficient learning and control in overactuated embodied systems.
>
> [Tan2025] Tan, C. K., Wang, C., Lyu, S., Hodossy, B. K., Schumacher, P., Wilson, E. B., … Song, S. (2025, May 12). Myoassist 0.1: Myosuite for dexterity and agility in bionic humans. 2025 International Conference On Rehabilitation Robotics (ICORR), 437–442.

---

> > ### Author Response · Authors · 2025-08-06
> >
> > Dear Reviewer ZrjQ:
> >
> > We would like to thank you again for your positive evaluation of our work. As the discussion period draws to a close, please let us know if there is any further clarification, analysis, or supporting material we could provide that might help to further strengthen the paper and address your concerns.

---

> ### Comment · Reviewer_ZrjQ · 2025-08-09
>
> I appreciate the authors' rebuttal, which addressed my concerns. Given this, I would like to keep my rating.

---

### Decision · Program_Chairs · 2025-09-18

**Decision:**

Accept (poster)

**Comment:**

This paper describes the MyoChallenge 2024, hold at NeurIPS 2024, targetting the control of simulated prosthesis using human limbs. The benchmark was divided into manipulation and locomotion tasks. An open-source simulation framework and standardized tasks were released for the challenge, that attracted significant participation.

As the reviewers report, the challenge is simplistic with respect to the long-term goal of recovering control over missing limbs, simulating only biomechanics and not the human neural system, and the entry barrier is very high, with a small percentage of teams being successful. However, the reviewers also acknowledge that the task is already enormously challenging, is a step in the right direction in a methodological and technical sense, and the task is well motivated and highly relevant. Simulation is well motivated, as large scale real experiments are even more complex in this domain. For this reasons, we recommend the acceptance of the paper.